# The Effects of *Agrimonia pilosa* Ledeb, *Anemone chinensis* Bunge, and *Smilax glabra* Roxb on Broiler Performance, Nutrient Digestibility, and Gastrointestinal Tract Microorganisms

**DOI:** 10.3390/ani12091110

**Published:** 2022-04-26

**Authors:** Rebekah L. McMurray, M. Elizabeth E. Ball, Mark Linton, Laurette Pinkerton, Carmel Kelly, Jonathan Lester, Caroline Donaldson, Igori Balta, Michael M. Tunney, Nicolae Corcionivoschi, Chen Situ

**Affiliations:** 1School of Biological Sciences, Queen’s University Belfast, Belfast BT9 5DL, UK; rmcmurray08@qub.ac.uk; 2Livestock Production Sciences Branch, Agri-Food and Biosciences Institute, Hillsborough BT26 6DR, UK; 3Bacteriology Branch Agri-Food and Biosciences Institute, Belfast BT9 5PX, UK; mark.linton@afbini.gov.uk (M.L.); laurette.pinkerton@afbini.gov.uk (L.P.); carmel.kelly@afbini.gov.uk (C.K.); igori.balta@gmail.com (I.B.); nicolae.corcionivoschi@afbini.gov.uk (N.C.); 4Devenish Nutrition, Belfast BT1 3BG, UK; jonny.lester@devenish.com (J.L.); Caroline.donaldson@delacon.com (C.D.); 5School of Pharmacy, Queen’s University Belfast, Belfast BT9 7BL, UK; m.tunney@qub.ac.uk; 6Institute for Global Food Security, Queen’s University Belfast, Belfast BT9 5DL, UK

**Keywords:** broiler chickens, plants, performance, lactic acid bacteria, *Agrimonia pilosa* Ledeb, *Anemone chinensis* Bunge, and *Smilax glabra* Roxb

## Abstract

**Simple Summary:**

There is a global effort to reduce the use of antibiotics in broiler production, and the antibacterial activity of extracts from *Agrimonia pilosa* Ledeb, *Anemone chinensis* Bunge, and *Smilax glabra* Roxb has been previously identified in vitro. The present study determines the effects of the dietary inclusion of these extracts on broiler production performance, nutrient digestibility, and on the levels of *Campylobacter* spp. and *Escherichia coli*, and lactic acid bacteria in the caecum. The inclusion of *S. glabra* Roxb and *A. chinensis* Bunge in broiler feed increased the number of lactic acid bacteria in the caeca of broilers, in comparison to the inclusion of antibiotics in poultry feed. The inclusion of *S. glabra* Roxb also reduced the levels of *E. coli* and *Campylobacter* spp., and improved the feed efficiency and weight gain. The plant extract of *A. chinensis* Bunge also increased weight gain in broilers. This study highlights the benefits of using *S. glabra* Roxb and *A. chinensis* Bunge in broiler feed as alternatives to antibiotics.

**Abstract:**

Poultry farming is growing globally, particularly in developing countries, to meet the demands of growing populations for poultry meat and eggs. This is likely to lead to an increase in the use of antibiotics in poultry feed, thus contributing to the development and spread of antibiotic resistance which, poses a serious threat to human and animal health worldwide. One way of reducing this threat is to reduce the use of antibiotics in poultry production by finding effective and sustainable antibiotic alternatives that can be used to support poultry health and productivity. Therefore, this study evaluates the incorporation of three medicinal plants, *Anemone chinensis* Bunge, *Smilax glabra* Roxb, and *Agrimonia pilosa* Ledeb, in poultry feed on production performance, nutrient digestibility, and bacteria in the chicken caecum in a 35-day performance trial with 420-day-old male Ross 308 broilers. Groups of randomly selected chicks received one of six dietary treatments. These included five experimental diets of reduced nutrient specifications as a negative control (NC); with amoxicillin as a positive antibiotic control (PC1); with *A. pilosa* Ledeb (NC1); with *A. chinensis* Bunge (NC2); and with *S. glabra* Roxb (NC3). One other positive control diet contained the recommended nutrient specification (PC2). Weight gain and feed intake were measured weekly and used to calculate the feed conversion ratio as performance parameters. Bacteria were enumerated from chicken caecum using a traditional plating method and selective agar. *S. glabra* Roxb and *A. chinensis* Bunge showed comparable effects to amoxicillin with significantly increased weight gain in birds offered these diets, compared to those offered the negative control from days 0 to 35 (*p* < 0.001). *S. glabra* Roxb exhibited effects similar to the amoxicillin control group with an improved feed conversion ratio (*p* < 0.001). In addition, *S. glabra* Roxb decreased numbers of *E. coli* and *Campylobacter* spp. on days 21 (*p* < 0.05) and 35 (*p* < 0.01) and increased numbers of lactic acid bacteria comparable to the antibiotic group on days 14 (*p* < 0.001) and 35 (*p* < 0.01). The findings of this in vivo trial highlight the potential of *S. glabra* Roxb and *A. chinensis* Bunge as beneficial feed material to promote poultry health and productivity in the absence of antibiotics.

## 1. Introduction

Poultry farming represents one of the most rapidly growing global markets within the agri-food industry. The nature of poultry farming with its relatively short production cycle, efficient feed conversion, and comparatively low costs per unit output have contributed to its worldwide growth [1]. In developing countries, the annual growth for global poultry meat production is estimated to increase from 2005 to 2050 by 2.4% [2]. Globally, poultry meat production is estimated to increase from 137 million tonnes in 2020 to 145 million tonnes in 2029 [3]. It is also predicted that this will contribute to one third of the overall 60% increase in the global consumption of antibiotics in animals raised for food from around 63,000 tons in 2010 to 104 000 tons by 2030 [4]. Many countries continue to use antibiotics in poultry feed for therapeutic use, prophylactic use, and to support growth [5], therefore the number of antibiotics used is likely to increase with the increasing demand for poultry meat. However, the increased use and misuse of antibiotics in poultry feed is one of the main causes of the development and spread of antibiotic resistance [6,7]. This increment in antibiotic usage could potentially cause increased bacterial resistance, which is associated with the emergence and spread of new types of multi-resistant foodborne bacteria, such as Salmonella, *Campylobacter* spp., and *Escherichia coli*, [8,9]. At a global level, one of the main strategies to reduce the growing threat of antibiotic resistance is to reduce, and ultimately phase out, the unnecessary use of antibiotics in animal farming [8,10,11]. This has stimulated increased interest in exploring alternatives to antibiotics in poultry feed [12]. Therefore, finding an effective and sustainable supplement that can be used to improve performance and bird health could contribute to reducing the amount of antibiotics used in poultry farming. Some research suggests that plants might be a viable alternative to antibiotics in poultry diets [13,14]. Studies of plant extracts used to supplement poultry diets have shown that they can provide similar benefits to antibiotics, such as inhibiting pathogens, increasing beneficial bacteria, and improving digestibility, growth and performance [15]. Therefore, the aim of the present study is to evaluate the effect of plant supplementation on broiler chicken performance, nutrient digestibility, and bacteria in the gut of broiler chickens in an in vivo poultry trial. The plants used in this study are *Agrimonia pilosa* Ledeb, *Anemone chinensis* Bunge, and *Smilax glabra* Roxb, which were selected based on the results from previous in vitro assays [16]. These three plant extracts exhibited significant in vitro antibacterial activity against *Listeria monocytogenes*, *Salmonella enteritidis*, and *Escherichia coli* over 24 h. It is hypothesised that the inclusion of the selected plants will improve broiler performance and increase beneficial bacteria in the caecum to levels equivalent to an antibiotic supplementation.

## 2. Materials and Methods

### 2.1. The Determination of Antibacterial Activity In Vitro

A reference strain of *Campylobacter jejuni* was obtained from the National Collection of Type Cultures (NCTC 11322). Clinical isolates of *C. jejuni* were obtained as frozen stocks from the Agri-Food and Biosciences Institute from retail packs of raw chicken (RC152, RC104, and RC179). A reference strain of *E. coli* was obtained from the American Type Culture Collection (ATCC 25922). Clinical isolates of *E. coli* were obtained as frozen stocks from Queen’s University Belfast initially from urinary tract infection sources (RC152, RC104, and RC179). Each bacterium was identified using 16S PCR, Sanger sequencing, and BLAST analysis as per manufacturer’s instructions for using MyTaq™ Red Mix [17]. *C. jejuni* and *E. coli* were selected for antimicrobial susceptibility testing because they are pathogens that cause foodborne diseases and are commonly found in the intestinal tract of poultry [18,19].

Three dry plant samples were purchased from a pharmaceutical company (Tong Ren Tang, Beijing) and included the herb of *Agrimonia pilosa* Ledeb, the tuber of *Smilax glabra* Roxb, and the root of *Anemone chinensis* Bunge. The accepted names of these plants are in accordance with a list available online [20]. These plants were prepared as in the previous methods [16].

The plant extracts were screened using the broth microdilution method [21] to determine the minimum inhibitory concentration (MIC) against *C. jejuni* and *E.coli*. Bacterial cultures were incubated overnight in conditions optimised for bacterial growth. *C. jejuni* was incubated in Mueller Hinton fastidious broth (MH-F) (Sigma-Aldrich, Oxoid, UK) at 41 ± 1 °C, microaerobic. *E.coli* was incubated in Mueller Hinton broth (MHB) (Sigma-Aldrich, Oxoid, UK) at 35 ± 1 °C [16].

### 2.2. Diets, Experimental Design, and Husbandry

Day-old male Ross 308 (*n* = 420) broiler chicks were randomly allocated to 30 pens (size 1.5 × 2 m). The trial was approved by the Animal Welfare Ethical Review Body at the Agri-Food and Biosciences Institute (AFBI) and conducted under the Animals Scientific Act 1986. The chickens were assigned to one of six dietary treatments (five pen replicates; 14 chicks per pen) and had free access to drinking water via bell drinkers. The temperature was gradually decreased from 32 °C to 22 °C. An 18 h light:6 h dark lighting programme was followed according to a commercial manual [22].

Birds were offered wheat/soyabean meal-based diets as mash (Table 1). The dietary treatments were the following: negative control (NC) (reduced nutrient specification), positive control (PC1) (NC + 40 mg/kg amoxicillin), PC2 (recommended nutrient specification), NC + herb of *A. pilosa* Ledeb (NC1), NC + root of *A. chinensis* Bunge (NC2), and NC + tuber of *S. glabra* Roxb (NC3). Commercial dry plants were purchased from Hutchison Whampoa Guangzhou Baiyunshan Chinese Medicine Co. Ltd. Each dried plant was powdered through a 1 mm screen using a hammer mill (Christie and Norris, AFBI) according to the manufacturer’s instructions, then added to diets at 20 g/kg. Starter (0–14 days) and grower/finisher (14–35 days) diets were formulated in line with breed recommendations [23] (Table 2) and offered ad libitum.

### 2.3. The Measurements and Sample Collection

Body weight was recorded weekly on a per pen, and individual basis and feed intake was recorded weekly on a per pen basis. Weight gain and the feed conversion ratio (FCR) were calculated. Mortality and morbidity rates were recorded throughout [24]. 

Litter samples were obtained to assess the litter quality at 20, 27, and 34 days from 6 evenly distributed positions within each pen, and mixed to create a composite sample per pen. The percentage moisture content was determined by weight loss [25] after sufficient oven drying (N175CF Genlab MINO/175/F) at 80 °C for 48h. Litter pH was determined as per manufacturer’s instructions and using a pH meter (Hanna Instruments, Fischer Scientific, Hampton, NH, USA). Ammonia concentrations from each pen were measured on days 20, 27, and 34. One 15 litre clear plastic container (470 mm × 300 mm × 170 mm) was placed in each pen. One Gastec Passive Dosi-tube No. 3D for ammonia (SKC, Dorset, UK), with a measurement range of 2.5 to 1000 ppm, was held in place in each plastic container with two level Diall M4 × 46 mm hollow wall anchor cup hooks for 4 h to obtain an on-the-spot time-weighted average. This is a modification by Devenish Nutrition Ltd. personnel to a previous method [26]. The ammonia concentrations were calculated as per manufacturer’s instructions. One NH_3_ monitor, the multi-gas detector eco (International Gas Detectors ltd, Stockport, UK) with a measurement range of 0–100 ppm, was also placed in each plastic container for an on-the-spot reading at 3 separate times at days 27 and 34.

On days 14 and 35, 2 broilers per pen were randomly selected, anaesthetised, then killed by cervical dislocation. The ileum was dissected from each bird [27], then immediately frozen. Ileal contents were dried in an oven (N175CF Genlab MINO/175/F) at 80 °C for 72 h. Ileal samples were powdered through a 1 mm screen using a hammer mill (Christie and Norris, AFBI), according to the manufacturer’s instructions. Ileal samples were combined on a per pen basis to produce six pen replicates of ileal contents per treatment.

On days 7, 14, 21, and 35, 5 birds from each treatment group were euthanised as above to determine the average bacterial count from the caecum. A 10^−1^ dilution was prepared by adding 1 g of the caecum contents to 9 mL of maximum recovery diluent (MRD) (Oxoid Code CM0733B Thermo Scientific, Waltham, MA, USA). A 10^−1^ to 10^−6^ dilution series was prepared in MRD. Lactic acid bacteria (LAB), *E. coli*, and *Campylobacter* spp. were enumerated by conventional microbiological techniques using a suitable selective agar. For LAB, a 1 mL portion of each dilution was pour-plated, in duplicate, using De Man, Rogosa and Sharpe (MRS ISO) agar (Oxoid Code CM1153B). Once set, the plates were overlaid with another layer of MRS agar to restrict oxygen and ensure an anaerobic incubation. To enumerate *E. coli*, 100 µL of each suitable dilution was spread-plated onto duplicate TBX agar plates (Oxoid Code CM0945B). For *Campylobacter* spp., 100 µL of each suitable dilution was spread plated onto mCCDA agar plates (Oxoid Code CM0739 supplemented with SR0155E). In addition, to reduce the limit of detection, 1 mL of the 10^−1^ dilution was spread over 3 mCCDA plates [28] and the plates were allowed to dry before incubation. The MRS agar plates were incubated at 30 °C for 72 h. The TBX plates were incubated at 37 °C for 24 h. The mCCDA plates were incubated in a microaerophilic workstation (Don Whitley Scientific, Bingley, UK) at 41.5 °C for 48 h, in an atmosphere of 5% oxygen, 10% carbon dioxide, and 85% Nitrogen.

The agar plates of an appropriate dilution level with 30 to 300 bacteria colonies were selected and colonies counted using a plate counter (Stuart Scientific, Bath, UK).

### 2.4. Laboratory Analysis

The dried samples of plants, diets, and ileal digesta were analysed for various chemical constituents, according to the methods outlined in AOAC [29]. All the results are reported on a DM basis. The gross energy (GE) content was measured using the isoperibol bomb calorimeter (Parr Instruments Co., Moline, IL, USA). Titanium dioxide content was measured Leone’s method [30] with some modifications [31]. Amino acid content was determined by ion-exchange chromatography and conducted by Scientec Ltd. according to methods of the European Commission [32]. 

### 2.5. Statistical Analysis and Calculations

One-way analysis of variance (ANOVA) was used to analyse the data using GraphPad Prism (Version 5.0). A value of *p* < 0.05 was set to determine the statistical significance. Pen was taken as the experimental unit. If a significant difference was found (*p* < 0.05), a Fisher’s Least Significant Difference test was run to test the pairwise differences between the treatment means. The feed conversion ratio was calculated as the ratio of weight of feed intake per pen to the average weekly weight gain per pen [33]. The FCR was adjusted for mortality by calculating the ratio of the weight of feed intake per pen to the average weekly weight gain of the survivors per pen + average weekly weight gain of the mortalities per pen [34]. The moisture content values were calculated using the following equation: percentage moisture content = [(Wet litter weight (g) − dry litter weight (g))/wet litter weight(g)] × 100 [35]. Ileal digestibility was calculated through the use of titanium dioxide as an indigestible marker [36]

## 3. Results

### 3.1. The Nutritional Composition of Plants

As shown in Table 3, *A. chinensis* Bunge contained the highest amino acid levels, ash, and oil levels, followed by *A. pilosa* Ledeb, and *S. glabra* Roxb with the exception of arginine, for which *S. glabra* Roxb contained the highest amount. Crude fibre content was highest for *A. pilosa* Ledeb (349 g/kg), followed by *A. chinensis* Bunge (280 g/kg), then *S. glabra* Roxb (258 g/kg).

### 3.2. Performance, Nutrient Digestibility, Litter Quality, and Mortality

From 0 to 35 days, diets containing *A. chinensis* Bunge (NC2), *S. glabra* Roxb (NC3), and amoxicillin (PC1) led to significantly increased weight gain (*p* < 0.001), compared to the weight gain of birds administered the other diets. This was reflected in the higher live weights at day 14 and day 35 for birds receiving diets containing amoxicillin (PC1) and *A. chinensis* Bunge (NC2). From 0 to 35 days, the feed conversion ratios were most efficient for the birds offered diets containing amoxicillin (PC1), *A. pilosa* Ledeb (NC1), and *S. glabra* Roxb (NC3), and least efficient in birds offered diets containing the reduced nutrient specification (NC) (*p* < 0.001; Table 4. From 0 to 35 days, the feed conversion ratios were more efficient in the birds fed diets containing the recommended nutrient specification (PC2), compared to the birds offered diets containing the reduced nutrient specification (NC) (*p* < 0.001; Table 5).

Overall, the diets containing *A. chinensis* Bunge (NC2) and *S. glabra* Roxb (NC3) resulted in similar improved growth rates and 35-day liveweights as birds offered the diets containing amoxicillin PC1), compared to the birds offered the reduced nutrient specification negative control diet (NC). In terms of the FCR, the diet containing *S. glabra* Roxb (NC3) improved the feed efficiency to a level equivalent to that achieved by birds offered the diet containing amoxicillin (PC1).

There were no significant differences in ammonia (NH_3_) output, litter moisture content. However, the litter in pens from birds receiving diets containing amoxicillin (PC1), the recommended nutrient specification (PC2), and *A. chinensis* Bunge (NC2) had significantly lower pH values than litter from birds receiving the reduced nutrient specification (NC), *A. pilosa* Ledeb (NC1), and *S. glabra* Roxb (NC3) (*p* < 0.001; Table 5).

The ileal digestibility of dry matter was higher in the recommended nutrient specification (PC2), compared to all other treatments (*p* < 0.01). The ileal digestibility of ash ranged from 0.24 for birds receiving the antibiotic (PC1) to the highest levels of 0.53 and 0.57 for birds offered the recommended nutrient specification (PC2) and the *A. chinensis* Bunge (NC2) supplemented treatment group (Table 6).

There was a total of 3.78% mortality. There were no significant differences in mortality between treatments and none of the mortality was due to treatments. Ten chickens died and seven chickens were euthanised as they were unable to stand or had breathing difficulties and associated with common causes of mortality.

### 3.3. Lactic Acid Bacteria, Campylobacter spp., and E. Coli

The inclusion of *S. glabra* Roxb in broiler diets, from days 14 to 21, reduced *Campylobacter* spp., caecal colonisation, at day 14, from 3.3 Log_10_ CFU g^−1^ to 2.2 Log_10_ CFU g^−1^ (*p* < 0.01; Figure 1). Similarly, *S. glabra* Roxb in the broiler diets decreased the detection of *E. coli* from 8.2 Log_10_ CFU g^−1^ to 7.4 Log_10_ CFU g^−1^ (*p* < 0.05; Figure 2).

Although the other treatment groups did not show significant differences between the numbers of pathogens, the trend in the results over a period of 0 to 35 days suggests that the diet containing the antibiotic (PC1) resulted in higher levels of *Campylobacter* spp. and *E. coli* than the other diets containing *A. pilosa* Ledeb (NC1)*, A. chinensis* Bunge (NC2), and *S. glabra* Roxb (NC3) (Figure 1 and Figure 2).

Our results also show that the recommended nutrient specification diet (PC2), the reduced nutrient specification diet (NC), *A. pilosa* Ledeb diet (NC1)*, A. chinensis* Bunge diet (NC2), and the *S. glabra* Roxb diet (NC3) resulted in significantly higher levels of lactic acid bacteria compared to amoxicillin (PC1) at day 14 (*p* < 0.001; Table 7). The negative control diet (NC) and the *S. glabra* Roxb (NC3) diet also resulted in significantly higher levels of lactic acid bacteria, in comparison to the levels detected when amoxicillin diet (PC1) was included in the diet at day 35 (*p* < 0.01; Table 7). These results suggest that, over the period of 0 to 35 days, the diets that did not contain the antibiotic resulted in significantly higher levels of lactic acid bacteria; in particular, the *A. chinensis* Bunge diet (NC2), and the *S. glabra* Roxb diet (NC3) resulted in the highest levels of lactic acid bacteria (Table 7).

## 4. Discussion

From the in vitro studies [16], the most effective antibacterial activity against *C. jejuni* and *E. coli* was exhibited by *A. pilosa* Ledeb. Following this, *A. chinensis* Bunge and *S. glabra* Roxb demonstrated strong antibacterial activity against *C. jejuni* and *E. coli*. A review of the literature suggested that crude extracts of *A. pilosa* Ledeb, *S. glabra* Roxb, and *A. chinensis* Bunge did not appear to have been previously screened for antibacterial activity against *C. jejuni*. These new findings highlighted the antibacterial activity of these medicinal plants against *C. jejuni* and justified their use in an in vivo poultry trial. The analysis of the nutritional chemical composition indicated that *A. chinensis* Bunge has the highest amino acid content, followed by *A. pilosa* Ledeb and *S. glabra* Roxb. Accurately formulating to amino acid requirements is known to improve performance [37]. There appears to be no previous research that demonstrates the nutritional chemical composition of these plants, and knowledge of the composition enables a more accurate formulation of the diets, as the inclusion rates of other ingredients can be modified accordingly to balance the diets for total amino acids and other essential nutrients. *S. glabra* Roxb contained the lowest crude protein content (4.6%) and amino acid content, compared to the other plants analysed in this study. The findings indicate substantial differences in plant nutritional value. They also highlight the need for analysis prior to dietary inclusion. It is recognised that, at the level of inclusion in this study, the nutritional contribution of these plants is minimal, but it is still of importance to accurately profile the nutritional profile of all ingredients to accurately formulate diets to the requirements.

The findings presented herein highlight that birds fed with diets containing *A. chinensis* Bunge and *S. glabra* Roxb have similar growth rates and live weights at day-35 to broilers fed diets that include amoxicillin. Other studies have also shown that the addition of antibiotics, such as salinomycin and zinc bacitracin [33,38], flavophospholipol antibiotic [39], and penicillin [40], to poultry diets has led to increased weight gain. However, this study has demonstrated that *A. chinensis* Bunge and *S. glabra* Roxb could be used in place of amoxicillin to result in similar growth weights, and therefore be an effective antibiotic alternative.

The addition of *A. pilosa* Ledeb, *A. chinensis* Bunge, and *S. glabra* Roxb to poultry diets resulted in improved bird growth rates. In terms of the FCR, the poultry diet that contained *S. glabra* Roxb resulted in comparable effects to amoxicillin and improved the feed efficiency of the birds, when compared to the birds in the negative control group. Previous studies have shown that birds fed diets supplemented with plants or plant extracts, or with antibiotics, can achieve an improved performance by the modification of the gastrointestinal microbiota [41]. The bird caecum has a functional role in nutrient absorption, which affects the overall performance and health [42]. Any increase in pathogens, such as *Campylobacter* spp. or *E. coli*, could have detrimental impacts on nutrient absorption, feed conversion ratios, and performance, whereas increases in lactic acid bacteria may have beneficial effects by improving growth performance, feed conversion efficiency, and immune responses, and help to combat enteric pathogens [43]. Our study shows that the addition of *amoxicillin* in broiler diets decreased the lactic acid bacteria in birds, highlighting that antibiotics can have a negative effect on beneficial bacteria, as well as harmful bacteria. Birds receiving diets containing amoxicillin showed fluctuations in the levels of lactic acid bacteria, *Campylobacter* spp., and *E. coli*. Birds on diets that contained amoxicillin initially showed reduced levels of lactic acid bacteria after starting the diet, and then they gradually increased until day 35. The gradual increase in lactic acid bacteria in birds on the diet that contained amoxicillin was lower than the levels of lactic acid bacteria found in birds on the other poultry diets. The findings in this study suggest that amoxicillin did not increase performance through selective effects towards beneficial bacteria, but rather through decreasing pathogens.

The changes in the levels of lactic acid bacteria in this study are similar to those reported in other poultry trials, which included antibiotics, monensin [44], zinc bacitracin [35], chlorotetracycline, virginiamycin, and amoxicillin [45]. These studies reported a decrease in lactic acid bacteria in the intestinal tract of poultry that were administered feed with antibiotics, as in this study. This study shows that diets containing *S. glabra* Roxb are more effective than diets containing amoxicillin in stimulating the development of lactic acid bacteria populations in the caecum by day 35. Other studies have also shown that one of the beneficial effects of including plants in the poultry diet is the occurrence of increased levels of lactic acid bacteria in the caecum [33,46]. This modulation of the caecum may have resulted in differences in the litter quality, but the results suggest that the litter moisture, pH, and ammonia output were not affected by the treatment.

The findings of the present study suggest that poultry diets containing *S. glabra* Roxb can decrease *E. coli* and *Campylobacter* spp., and increase lactic acid bacteria. This is strong evidence to support the concept of plant-extract-based alternatives to antibiotics in broiler production. Other research has also shown that adding plant supplements, such as oregano oil, rosemary, eugenol, and carvacrol, can reduce *E. coli* and *C. jejuni* in the broiler caecum [3,24,47,48,49] and increase the abundance of *Lactobacillus* spp. [33], thus positively modulating the microbiota in the caecum. However, it is important to stress that the active agent in each of these plant supplements differs, and this points to the need for further work to identify the mode-of-action and active agent of the plants used in the present study. The modulation of the GI tract microbiota is commonly associated with improved digestibility, which can lead to improved growth and performance [50], but this was not observed in the present study. In this study, the ileal digestibility of dry matter was highest in the birds offered the recommended nutrient specification diet (PC2), compared to that of birds offered other treatments (Table 7). This could be explained due to the absence of and reduction in rapeseed meal and wheat pollard in the recommended nutrient specification diet (PC2), which was necessary to formulate to the higher specification. Rapeseed meal and wheat pollard were used to reduce the nutrient content of the diet, thus providing a nutritional challenge to the birds offered the negative control to allow for a response to the additives to be observed. These components contain more fibrous components, which leads to decreased digestibility [51]. The fact that ileal digestibility of dry matter was higher in the PC control diets is a reflection of the nutrient specification, and thus indicates that the mechanism of action for improved performance in this study is more likely due to the modulation of the microbiota, rather than nutrient utilisation.

## 5. Conclusions

The present study shows that the inclusion of plants in poultry feed can contribute to the control of pathogenic bacteria (*Campylobacter* spp. and *E. coli*), while enhancing beneficial bacteria (such as lactic acid bacteria) in the intestinal tract of poultry. The findings highlight the potential of using medicinal plants in feed to promote poultry health and performance in the absence of antibiotics, therefore contributing to the global efforts of reducing the consumption of antibiotics in livestock farming. The in vivo poultry study demonstrated that the inclusion of *S. glabra* Roxb and *A. chinensis* Bunge in broiler feed increased the number of lactic acid bacteria in the caeca of broilers, in comparison to the inclusion of antibiotics in poultry feed. Furthermore, the inclusion of *S. glabra* Roxb reduced the levels of *E. coli* and *Campylobacter* spp., and improved the feed conversion ratio and weight gain to levels comparable to the antibiotic supplemented. The plant extract of *A. chinensis* Bunge also increased weight gain in the broilers. This study highlights the benefits of using *S. glabra* Roxb and *A. chinensis* Bunge in broiler feed as alternatives to antibiotics.

## Figures and Tables

**Figure 1 animals-12-01110-f001:**
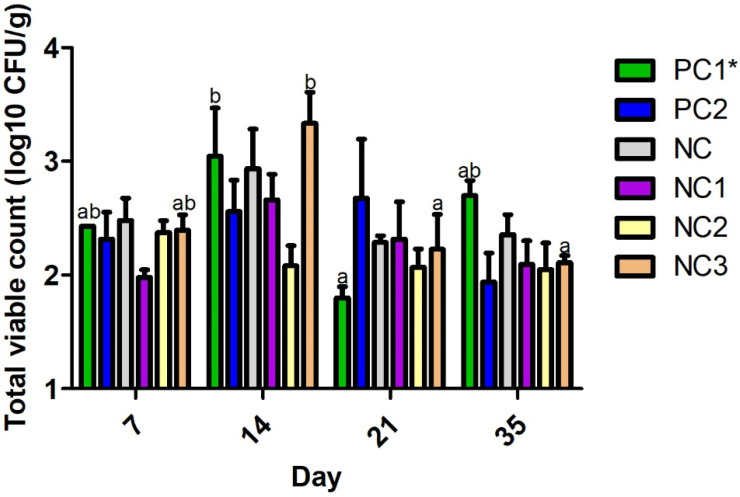
The effect of the plant materials on the bacterial load of *Campylobacter* spp. in the caecum from 7 to 35 days of age. Means without common superscripts (^ab^) indicate the significant differences (*, *p* < 0.05)between the treatments. PC1 = positive control 1 (NC+40mg/kg amoxicillin); PC2 = positive control 2 (recommended nutrient specification); NC = negative control (reduced nutrient specification); NC1 = NC + herb of *A. pilosa* Ledeb; NC2 = NC + herb of *A. chinensis* Bunge; and NC3 = NC + tuber of *S. glabra* Roxb.

**Figure 2 animals-12-01110-f002:**
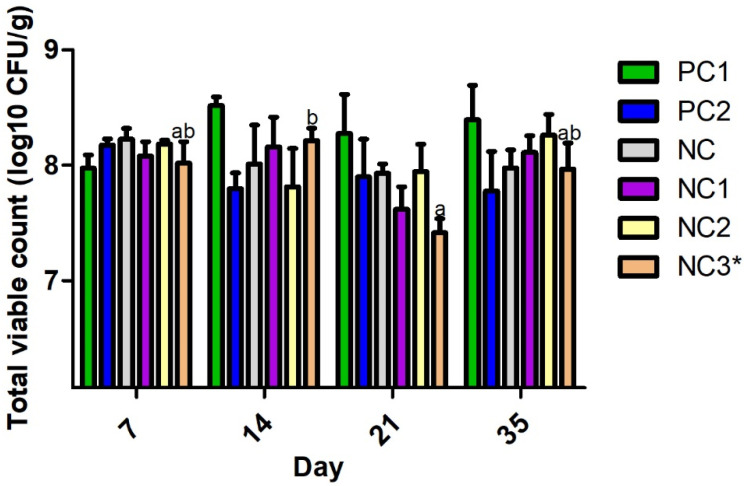
The effect of the plant materials on the bacterial load of *E. coli* in the caecum from 7 to 35 days of age. Means without common superscripts (^ab^) indicate the significant differences (*, *p* < 0.05) between the treatments. PC1 = positive control 1 (NC + 40mg/kg amoxicillin); PC2 = positive control 2 (recommended nutrient specification); NC = negative control (reduced nutrient specification); NC1 = NC + herb of *A. pilosa* Ledeb; NC2 = NC + herb of *A. chinensis* Bunge; and NC3 = NC + tuber of *S. glabra* Roxb.

**Table 1 animals-12-01110-t001:** Composition (g/kg) of the experimental treatment diets.

Feed Ingredient	Starter (0–14 d)						Grower/Finisher (14–35 d)					
	PC1	PC2	NC	NC1	NC2	NC3	PC1	PC2	NC	NC1	NC2	NC3
Wheat	640	650	640	619	619	619	626	678	626	612	612	612
Wheat pollard	50	0	50	50	50	50	100	25	100	100	100	100
Rapeseed meal	30	0	30	25	25	25	25	0	25	0	0	0
Soybean meal 48	214	252	214	220	220	220	180	180	180	195	195	195
Full fat soya	0	20	0	0	0	0	0	54	0	10	10	10
Soy oil	20	30	20	20	20	20	30	30	30	25	25	25
Salt	5	5	5	5	5	5	1	1	1	1	1	1
Sodium bicarbonate	5	5	5	5	5	5	5	0	5	5	5	5
DL methionine	2	2	2	2	2	2	1.8	1.9	1.8	1.8	1.8	1.8
Lysine HCl	3.7	3.0	3.7	3.6	3.6	3.6	3.2	2.7	3.2	2.9	2.8	2.9
Threonine	1.2	1.0	1.2	1.2	1.2	1.2	1.4	1.3	1.4	1.3	1.3	1.3
Limestone	2	4	2	2	2	2	1	1	1	1	1	1
Mono-dicalcium phosphate	9.5	10	9.5	9.5	9.5	9.5	7.5	7.5	7.5	7.4	7.4	7.4
Vitamin premix *	15	15	15	15	15	15	15	15	15	15	15	15
Titanium dioxide	3	3	3	3	3	3	3	3	3	3	3	3
Amoxicillin 10 mg/kg	0.1	0	0	0	0	0	0.1	0	0	0	0	0
*A. chinensis* Bunge	0	0	0	0	20	0	0	0	0	0	20	0
*S. glabra* Roxb	0	0	0	0	0	20	0	0	0	0	0	20
*A. pilosa* Ledeb	0	0	0	20	0	0	0	0	0	20	0	0

PC1 = positive control 1 (NC + 40mg/kg amoxicillin); PC2 = positive control 2 (recommended nutrient specification); NC = negative control (reduced nutrient specification); NC1 = NC + herb of *A. pilosa* Ledeb; NC2 = NC + herb of *A. chinensis* Bunge; and NC3 = NC + tuber of *S. glabra* Roxb. * Starter supplied per tonne of diet: vitamin A 10MIU; vitamin D3 5MIU; vitamin E 100 g; vitamin K 4 g; thiamine (B1) 3 g; riboflavin (B2) 6 g; pyridoxine 5 g; vitamin B12 27.5 mg; biotin (2.5%) 0.25 g; calcium pantothenate 12.5 g; nicotinic 45 g; folic acid 1.5 g; iodate-calcium 2 g; selenite-sodium 0.25 g; iron sulphate 45 g; molybdate-sodium 0.5 g; manganese oxide 90 g; copper sulphate 15 g; zinc oxide 90 g; betaine 500 g; optiphos 5000 25 g; and hostazyme 50 g.

**Table 2 animals-12-01110-t002:** Formulated analysis of the treatment diets (digestible amino acids) (fresh basis).

Nutrient (g/kg)	Starter (0–14 d)						Grower/Finisher (14–35 d)					
	PC1	PC2	NC	NC1	NC2	NC3	PC1	PC2	NC	NC1	NC2	NC3
Dry matter	879	879	878	889	886	883	891	887	892	895	893	914
Crude protein	209	219	209	209	209	208	194	202	194	196	196	195
Metabolisable energy MJ/kg	12.84	13.17	12.84	12.89	12.87	12.88	13.17	13.56	13.17	13.30	13.29	13.30
Calcium	9.5	9.5	9.5	9.5	9.5	9.5	7.9	7.8	7.9	7.8	7.8	7.8
Phosphorus	6.9	7.0	6.9	6.8	6.8	6.8	6.3	6.3	6.3	6.2	6.2	6.2
Available phosphorus	4.4	4.5	4.4	4.4	4.4	4.4	4.0	4.0	4.0	4.0	4.0	4.0
Crude fat	35.3	47.6	35.3	35.3	35.3	35.0	44.8	54.0	44.8	41.9	41.9	41.6
Crude fibre	27.1	25.0	27.1	33.3	31.9	31.4	26.1	25.7	26.1	31.0	29.7	29.2
Methionine	5.1	5.2	5.1	5.1	5.1	5.1	4.7	4.9	4.7	4.7	4.7	4.7
Cysteine	3.8	3.8	3.8	3.7	3.7	3.7	3.6	3.6	3.6	3.5	3.5	3.5
Methionine + cysteine	9.0	9.1	9.0	9.0	9.0	8.9	8.4	8.6	8.4	8.3	8.3	8.3
Lysine	12.5	12.9	12.5	12.5	12.5	12.4	11.0	11.3	11.0	11.1	11.0	11.0
Histidine	5.1	5.4	5.1	5.1	5.1	5.1	4.7	4.9	4.7	4.7	4.7	4.7
Tryptophan	2.5	2.7	2.5	2.5	2.5	2.5	2.3	2.4	2.3	2.3	2.3	2.3
Threonine	8.4	8.6	8.4	8.4	8.4	8.3	7.9	8.2	7.9	7.9	7.9	7.9
Arginine	12.6	13.6	12.6	12.6	12.6	12.6	11.4	12.2	11.4	11.7	11.7	11.7
Isoleucine	8.0	8.7	8.0	8.1	8.1	8.0	7.3	7.8	7.3	7.5	7.5	7.5
Leucine	14.7	15.7	14.7	14.7	14.7	14.6	13.5	14.3	13.5	13.8	13.8	13.7
Phenylalanine	9.6	10.3	9.6	9.6	9.6	9.6	8.8	9.5	8.8	9.0	9.0	9.0
Tyrosine	6.7	7.2	6.7	6.7	6.7	6.6	6.1	6.5	6.1	6.2	6.2	6.2
Valine	9.2	9.7	9.2	9.2	9.2	9.1	8.5	8.9	8.5	8.6	8.6	8.5
Glycine	8.4	8.8	8.4	8.4	8.4	8.3	7.7	8.1	7.7	7.8	7.8	7.7
Serine	9.4	10.1	9.4	9.4	9.4	9.4	8.6	9.2	8.6	8.8	8.8	8.8
Phenylalanine + tyrosine	16.2	17.5	16.2	16.3	16.3	16.2	14.9	16.0	14.9	15.2	15.3	15.2

PC1 = positive control 1 (NC + 40 mg/kg amoxicillin); PC2 = positive control 2 (recommended nutrient specification); NC = negative control (reduced nutrient specification); NC1 = NC + herb of *A. pilosa* Ledeb; NC2 = NC + herb of *A. chinensis* Bunge; and NC3 = NC + tuber of *S. glabra* Roxb.

**Table 3 animals-12-01110-t003:** Nutritional composition of the dried plants as they arrived commercially dried.

Nutritional Compound	Total Composition g/kg		
	*A. pilosa* Ledeb	*S. glabra* Roxb	*A. chinensis* Bunge
Crude protein (N × 6.25) (Dumas)	72	46	91
Crude fibre	349	258	280
Ash	70	20	120
Total oil (oil B)	19	4.2	20.1
Alanin	3.0	0.9	3.9
Arginine	3.2	4.4	4.0
Aspartic acid	6.7	2.7	9.3
Cysteine	0.8	0.7	1.0
Glutamic acid	6.8	2.9	8.9
Glycine	3.4	1.4	4.1
Histidine	1.4	0.5	1.6
Iso-leucine	2.9	1.0	3.8
Leucine	5.0	1.7	6.5
Lysine	3.6	1.5	4.0
Methionine	1.3	0.5	1.5
Phenylalanine	3.5	1.1	4.1
Proline	2.8	1.1	4.4
Serine	3.1	1.6	3.9
Threonine	3.0	1.4	3.8
Tyrosine	1.9	0.8	2.4
Valine	0.36	0.15	0.46
Tryptophan	0.09	0.03	0.11
Gross energy (MJ/kg)	16.32	16.04	15.50

PC1 = positive control 1 (NC + 40 mg/kg amoxicillin); PC2 = positive control 2 (recommended nutrient specification); NC = negative control (reduced nutrient specification); NC1 = NC + herb of *A. pilosa* Ledeb; NC2 = NC + herb of *A. chinensis* Bunge; and NC3 = NC + tuber of *S. glabra* Roxb.

**Table 4 animals-12-01110-t004:** The effect of the plant materials on feed intake (FI), weight gain (WG), and feed conversion ratio (FCR) from 0 to 35 days of age.

Day	Parameter	Treatment						SEM	*p*-Value
		PC1	PC2	NC	NC1	NC2	NC3		
0–14	FI (g)	581.2 ^bc^	510.0 ^a^	542.8 ^ab^	551.9 ^ab^	605.8 ^c^	571.9 ^bc^	16.72	<0.01
	WG (g)	493.8 ^b^	433.0 ^a^	437.6 ^a^	465.5 ^ab^	483.0 ^b^	460.3 ^ab^	10.85	<0.01
	FCR	1.18	1.18	1.24	1.19	1.25	1.24	0.024	NS
14–35	FI (g)	2624.6 ^a^	2562.3 ^a^	2540.1 ^a^	2465.6 ^a^	2856.2 ^b^	2626.1 ^ab^	81.72	<0.05
	WG (g)	1737.7 ^b^	1556.6 ^a^	1450.1 ^a^	1566.1 ^a^	1757.6 ^b^	1776.4 ^b^	39.13	<0.001
	FCR	1.51^a^	1.65 ^bc^	1.75 ^c^	1.58 ^ab^	1.63 ^b^	1.48 ^a^	0.035	<0.001
0–35	FI (g)	3205.9 ^ab^	3072.3 ^a^	3082.9 ^a^	3017.5 ^a^	3462.0 ^b^	3197.7 ^a^	86.99	<0.05
	WG (g)	2231.5 ^b^	1989.6 ^ab^	1887.7 ^a^	2031.6^b^	2240.6 ^b^	2236.7 ^b^	41.96	<0.001
	FCR	1.44 ^a^	1.54 ^b^	1.63 ^c^	1.49 ^ab^	1.55 ^b^	1.43 ^a^	0.027	<0.001
0	Live weight (g)	44.14	43.82	43.8	43.42	44.46	43.54	0.385	NS
14		538.0 ^c^	476.8 ^a^	481.4 ^a^	508.9 ^abc^	527.5 ^bc^	503.8 ^ab^	10.94	<0.01
35		2275.7 ^b^	2033.4 ^ab^	1931.5 ^a^	2075.0 ^b^	2285.0 ^c^	2280.2 ^c^	41.95	<0.001

Means without common superscripts (^abc^) indicate the significant differences between the treatments (rows). PC1 = positive control 1 (NC + 40 mg/kg amoxicillin); PC2 = positive control 2 (recommended nutrient specification); NC = negative control (reduced nutrient specification); NC1 = NC + herb of *A. pilosa* Ledeb; NC2 = NC + herb of *A. chinensis* Bunge; and NC3 = NC + tuber of *S. glabra* Roxb. NS = non-significant (*p* > 0.05).

**Table 5 animals-12-01110-t005:** The effect of the plant materials on the litter characteristics at different stages of production.

Day	Parameter	PC1	PC2	NC	NC1	NC2	NC3	SEM	*p*-Value
20	Ammonia dosi (ppm)	11.7	10.8	11.3	18.9	10.3	13.10	3.423	NS
	pH	8.24	8.02	7.88	7.74	8.025	8.040	0.408	NS
	Moisture content (%)	39.7	41.7	43.1	39.6	38.0	37.8	4.13	NS
27	Ammonia dosi (ppm)	15.5	26.0	9.5	21.0	21.0	19.00	6.24	NS
	Ammonia monitor (ppm)	9.7	7.0	8.8	19.0	9.7	13.78	5.909	NS
	pH	8.12	7.58	8.64	8.24	7.64	8.340	0.314	NS
	Moisture content (%)	48.6	51.7	45.4	52.2	51.2	50.5	4.37	NS
34	Ammonia dosi (ppm)	42.0	53.5	47.5	64.0	40.5	50.0	14.08	NS
	Ammonia monitor (ppm)	20.1	27.8	25.1	28.8	21.7	21.3	4.82	NS
	pH	6.56 ^a^	7.48 ^ab^	8.34 ^bc^	8.76 ^c^	7.14 ^a^	8.50 ^bc^	0.351	<0.001
	Moisture content (%)	55.0	62.70	56.0	62.8	61.7	61.0	3.03	NS

Means without common superscripts (^abc^) indicate the significant differences between the treatments (rows). PC1 = positive control 1 (NC+40mg/kg amoxicillin); PC2 = positive control 2 (recommended nutrient specification); NC = negative control (reduced nutrient specification); NC1 = NC + herb of *A. pilosa* Ledeb; NC2 = NC + herb of *A. chinensis* Bunge; and NC3 = NC + tuber of *S. glabra* Roxb. NS = non-significant (*p* > 0.05).

**Table 6 animals-12-01110-t006:** The effect of the plant materials on the apparent ileal digestibility (%).

Digestibility	PC1	PC2	NC	NC1	NC2	NC3	SEM	*p*-Value
Dry matter (%)	66 ^a^	79 ^b^	66 ^a^	70 ^a^	67 ^a^	70 ^a^	0.022	<0.01
Ash (%)	24 ^a^	53 ^c^	48 ^c^	38 ^b^	57 ^c^	36 ^b^	0.035	<0.001
Crude protein (%)	76	83	73	79	76	77	0.022	NS

Means without common superscripts (^abc^) indicate the significant differences between the treatments (rows). PC1 = positive control 1 (NC + 40 mg/kg amoxicillin); PC2 = positive control 2 (recommended nutrient specification); NC = negative control (reduced nutrient specification); NC1 = NC + herb of *A. pilosa* Ledeb; NC2 = NC + herb of *A. chinensis* Bunge; and NC3 = NC + tuber of *S. glabra* Roxb. NS = non-significant (*p* > 0.05).

**Table 7 animals-12-01110-t007:** The effects of the treatment groups on the caecum lactic acid bacteria bacterial load at 14 and 35 days of age.

Treatment Group	Day	
	14	35
	Lactic Acid Bacteria (log_10_ CFU g^−1^)	
PC1	3.68 ^a^	7.80 ^a^
PC2	9.16 ^b^	8.65 ^ab^
NC	10.05 ^b^	9.30 ^c^
NC1	9.69 ^b^	8.61 ^ab^
NC2	9.54 ^b^	8.93 ^b^
NC3	9.40 ^b^	9.64 ^c^
SEM	0.339	0.308
*p*-value	<0.001	<0.01

Means without common superscripts (^abc^) indicate the significant differences between the treatments (rows). PC1 = positive control 1 (NC + 40 mg/kg amoxicillin); PC2 = positive control 2 (recommended nutrient specification); NC = negative control (reduced nutrient specification); NC1 = NC + herb of *A. pilosa* Ledeb; NC2 = NC + herb of *A. chinensis* Bunge; and NC3 = NC + tuber of *S. glabra* Roxb.

## Data Availability

The data presented in this study are available on request from the corresponding author.

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
