# Peer review of "The Effects of Agrimonia pilosa Ledeb, Anemone chinensis Bunge, and Smilax glabra Roxb on Broiler Performance, Nutrient Digestibility, and Gastrointestinal Tract Microorganisms"

_animals, 2022, doi:10.3390/ani12091110_

Round 1

Reviewer 1 Report

In my evaluation, this manuscript is prepared very well with proper method. However, some of minor comments must be clarified before publication. Please find my comments below:

Main comments

  1. Please state your hypothesis in the Introduction
  2. Please add the reason why the authors used the plant extracts from A. pilosa Ledeb, A. chinensis Bunge, S. glabra Roxb instead of another plant extracts in this study. What make those plant extract so special? These information will be better if mentioned in the introduction.
  3. L207-208 Why MRS was incubated aerobically? Why not anaerobically? As LAB medium, it should be incubated in anaerobe.
  4. Why the authors didn’t use statistical analysis in the Table 4?
  5. What kind of antibacterial substrate contained in these plant extracts? Did the author analyze it?
  6. Please add the description of every abbreviations in the footnote of tables and figures, including abbreviation of treatments.

Minor comments

L 70-72 ‘Globally,…. tonnes in 2020 [3]’. Now is 2022, please add the update reference.

L94      ‘A. pilosa Ledeb, A. chinensis Bunge, S. glabra Roxb’. Please write those words in the full name as first mention in the main text.

L99      ‘C.jejuni’. Please write in the full name as first mention in the main text.

L110    ‘Tong Ren Tang’. Is this a Pharmaceutical company? If yes, please mention it in the text.

The use of ‘oC’ in the some sentences is not consistent. Please check and revise it.

L123    ‘AFBI’. Please write in the full name as first mention in the main text.

L 198   ‘Escherichia coli’. The author can write as E. coli since it was already written as the full name in the previous sentence.

Please change ‘ml’ to ‘mL’ in the text.

Please change ‘µl’ to ‘µL’ in the text.

L216    ‘AOAC (1990). In my opinion author can refer the updated AOAC.

Please add the description for ‘NS’ in the footnote of every tables

The authors redundant many time of treatment abbreviation in the Results.  In my opinion, the author can describe the treatment abbreviation in the footnote of each table and figure. Therefore in the text of Results, the author can directly use the abbreviation without any long description.

Please check the footnote for explain superscript. Some of Table have superscript not only ‘a’ and ‘b’, but also ‘c’

In Table 2, the analysis is based on dry matter or as feed/fresh feed? It will be better if the data was presented as based on dry matter content.

Author Response

Thank you for your comments and suggestions. Please below the response to your comments.

Main comments:

  1. Please state your hypothesis in the Introduction

A hypothesis has now been included at the end of the introduction. 

  1. Please add the reason why the authors used the plant extracts from A. pilosa Ledeb, A. chinensis Bunge, S. glabra Roxb instead of another plant extracts in this study. What make those plant extract so special? These information will be better if mentioned in the introduction.

These extracts were chosen because they exhibited significant antibacterial activity in vitro and in a poultry digest model in previous work by our group (McMurray et al 2020, ref 16). Some more details have been added to the introduction. 

  1. L207-208 Why MRS was incubated aerobically? Why not anaerobically? As LAB medium, it should be incubated in anaerobe.

It was incubated in an aerobic incubator. However during preparation MRS agar was poured into the agar plate. The caecum solution was pipetted onto the agar. Following this, a layer of MRS agar was added on top to starve/restrict the oxygen available to the LAB and hence incubated in the anaerobe. The text in the manuscript has been modified to make this clearer.

  1. Why the authors didn’t use statistical analysis in the Table 4?

Based on Reviewer 2 recommendations, Table 4 has been removed from the manuscript.

  1. What kind of antibacterial substrate contained in these plant extracts? Did the author analyze it?

The aim of this study was to evaluate in vivo effects of these plants in poultry production after the in vitro antibacterial testing. Analysing antibacterial substances contained in these extracts as supporting evidence may be carried out now that the in vivo effects have been established and to support commercial application. Observed effects, including the antibacterial effect is a result of the multitargeting effects of different phytochemicals contained in the plant/extract, which is fundamentally different from the single-targeted antibiotics.

  1. Please add the description of every abbreviations in the footnote of tables and figures, including abbreviation of treatments.

Abbreviations have now been included in tables and figures

Minor comments: 

  1. L 70-72 ‘Globally,…. tonnes in 2020 [3]’. Now is 2022, please add the update reference.

Reference updated in the text.

  1. L94      ‘A. pilosa Ledeb, A. chinensis Bunge, S. glabra Roxb’. Please write those words in the full name as first mention in the main text.

Now in full at first mention.

  1. L99      ‘C.jejuni’. Please write in the full name as first mention in the main text.

Now in full at first mention.

  1. L110    ‘Tong Ren Tang’. Is this a Pharmaceutical company? If yes, please mention it in the text.

Yes, it is a pharmaceutical company which is now included in the text.

  1. The use of ‘oC’ in the some sentences is not consistent. Please check and revise it.

Have been retyped.

  1. L123    ‘AFBI’. Please write in the full name as first mention in the main text.

In full at first mention.

  1. L 198   ‘Escherichia coli’. The author can write as E. colisince it was already written as the full name in the previous sentence.

This has been changed. 

  1. Please change ‘ml’ to ‘mL’ in the text.

Have been changed

  1. Please change ‘µl’ to ‘µL’ in the text.

Have been changed.

  1. L216    ‘AOAC (1990). In my opinion author can refer the updated AOAC.

This has been updated.

  1. Please add the description for ‘NS’ in the footnote of every tables

Description added to footnotes.

  1. The authors redundant many time of treatment abbreviation in the Results.  In my opinion, the author can describe the treatment abbreviation in the footnote of each table and figure. Therefore in the text of Results, the author can directly use the abbreviation without any long description.

The authors believe that the descriptions in the text are necessary to aid the understanding of the results.

  1. Please check the footnote for explain superscript. Some of Table have superscript not only ‘a’ and ‘b’, but also ‘c’

"C" has been added to the footnotes 

  1. In Table 2, the analysis is based on dry matter or as feed/fresh feed? It will be better if the data was presented as based on dry matter content.

The data is on a fresh basis. This is the normal presentation for formulations in nutrition work as diets are formulated on a fresh basis. 

Reviewer 2 Report

The choice of experimental treatments is not clear in the manuscript.  The reason for using reduced nutrient concentrations needs to be explained. It would have been more interesting to test the PC2 with and without amoxi vs a commercial diet with the added medicinal plants.  It would be more commercially relevant. 

The paper needs more focus. For example, the results of plant composition are not relevant as the plants were not included in the formulation as ingredients contributing nutrients, but rather at a dosage for their medical properties. Similarly, the results of in vitro microbial activity are not related to the aims of the study.  They should be reported in a different manuscript.

Author Response

Thank you for your comments and suggestions. Please below the response to your comments.

The choice of experimental treatments is not clear in the manuscript.  The reason for using reduced nutrient concentrations needs to be explained. It would have been more interesting to test the PC2 with and without amoxi vs a commercial diet with the added medicinal plants.  It would be more commercially relevant. 

There are two ways of evaluating alternatives to antibiotics; the first being the way it was done in this manuscript in testing them in a more challenged situation (i.e. with reduced nutrients) and the other being how you describe the treatments +/- with a normal diet. The treatments you suggest were considered in the trial design but based on previous work (by us and others), when trials are conducted in a research environment with optimum husbandry and environmental conditions, there is often not a response to antibiotics or alternatives unless the birds are challenged in some way. In this case, we chose to challenge the birds with a diet somewhat below nutritional requirements so as an effect with the antibiotic (and potentially with the alternatives) could be observed…i.e. give space for a response.

The paper needs more focus. For example, the results of plant composition are not relevant as the plants were not included in the formulation as ingredients contributing nutrients, but rather at a dosage for their medical properties. Similarly, the results of in vitro microbial activity are not related to the aims of the study.  They should be reported in a different manuscript.

Table 4 containing the in vitro microbial activity has been removed. The authors feel it is important to retain the nutritional profile of the plants as this data is extremely limited in the literature and it adds to the portfolio of knowledge on the plants. We acknowledge that the plants are not making a significant nutritional contribution and have made this clear in the text.

Reviewer 3 Report

   I read the revised version of manuscript entitled “Effects of Agrimonia pilosa Ledeb, Anemone chinensis Bunge, and Smilax glabra Roxb on broiler performance, nutrient digestibility and gastrointestinal tract microorganisms” for possible publication in Animals. The experimental design, methods and conducting are appropriate, but the statistical analyses were not done enough and, therefore, the summarized results are clear. I recommend that the manuscript should be revised largely for the following points:

 1)  The statistical analyses were not done enough. The authors should carry out multiple comparisons of the means in treatment groups by a multiple range test, such as tukey’s test, after ANOVA. And then, for counting of Campylobacter, E. coli and Lactic acid bacteria in caecum, comparisons of the means between sampling age are also required. After that, text of “Results” in L236-328 and “Discussion” in 344-434 should be completely written again.

 2) Probably, it is not correct that in “Discussion” the authors lumped various materials derived from plants together, such as agrimonia pilosa, anemone chinensis, smilax glabra, oregano, rosemary, moringa oleifera and citrus, as similar bioactive materials for antibacterial function or modulating the microbiota. The various materials contain different active components, respectively. The authors should give each one more careful consideration.

 3) The formatting of the cited papers should be applied to the Animals format. For example, “on [16]” in L119 should be changed to “on previous study [16]”; “(Aviagen, 2019; Table 2)” in L135-136 should be changed to “[23] (Table 2)”; “AOAC (1990)” in L216 should be changed to “AOAC [29]”; “Leone (1973) and modified by [31]” in L219 should be changed to “Leone’s method [30] with some modifications [31]”; “[32]” should be inserted between “Standardisation” and “namely” in L221; “Commission (2009)” in L222 should be changed to “Commission [33]”; “institute (2005)” in L223 should be changed to “institute [34]”. And then, what does “(56)” in L430 means? On the other hand, we cannot find out “[54]” of the References list in the text of this manuscript.

 4) Others: “and in Campylobacter” in L30-31 may be wrong in English; “liveweight, liveweight gain, mortality” in L273 should be deleted, because there are no data of those parameters in Table 6 and the results in mortality were described elsewhere (in L283-286); The authors said that the plant extract in this study increased lactic acid bacteria in L391-393, but I think that the bacteria were simply increased by dietary condition without amoxicillin from the results in this study.; Lines of “Phosphorus”, “Crude Fat”, “Crude Fibre”, “Methionine”, “Cysteine”, “Histidine”, “Arginine”, “Isoleucine”, “Leucine”, “Phenylalanine”, “Tyrosine”, “Valine”, “Glycine”, “Serine” and “Phenylalanine + Tyrosine” in Table 2 should be omitted, because those nutrients are not important in this study. And then, Table 1 and 2 should be combined into one table.

Author Response

Thank you for your comments and suggestions. Please below the response to your comments.

I read the revised version of manuscript entitled “Effects of Agrimonia pilosa Ledeb, Anemone chinensis Bunge, and Smilax glabra Roxb on broiler performance, nutrient digestibility and gastrointestinal tract microorganisms” for possible publication in Animals. The experimental design, methods and conducting are appropriate, but the statistical analyses were not done enough and, therefore, the summarized results are clear. I recommend that the manuscript should be revised largely for the following points:

  • The statistical analyses were not done enough. The authors should carry out multiple comparisons of the means in treatment groups by a multiple range test, such as tukey’s test, after ANOVA. And then, for counting of Campylobacter, E. coli and Lactic acid bacteria in caecum, comparisons of the means between sampling age are also required. After that, text of “Results” in L236-328 and “Discussion” in 344-434 should be completely written again.

Apologies – the description in the statistical analysis and calculations section was incomplete in the manuscript. Fisher’s LSD test was ran to test for pairwise differences. The text has been updated to state this.

The authors believe that a comparison between ages is not of value because we will not take in account the major differences in microbiome at different stages in bird’s life. The aim of the trial was to investigate differences between experimental treatments and not examine the effect of age on the microbiome.

 2) Probably, it is not correct that in “Discussion” the authors lumped various materials derived from plants together, such as agrimonia pilosa, anemone chinensis, smilax glabra, oregano, rosemary, moringa oleifera and citrus, as similar bioactive materials for antibacterial function or modulating the microbiota. The various materials contain different active components, respectively. The authors should give each one more careful consideration.

The text has been modified accordingly.                               

 3) The formatting of the cited papers should be applied to the Animals format. For example, “on [16]” in L119 should be changed to “on previous study [16]”; “(Aviagen, 2019; Table 2)” in L135-136 should be changed to “[23] (Table 2)”; “AOAC (1990)” in L216 should be changed to “AOAC [29]”; “Leone (1973) and modified by [31]” in L219 should be changed to “Leone’s method [30] with some modifications [31]”; “[32]” should be inserted between “Standardisation” and “namely” in L221; “Commission (2009)” in L222 should be changed to “Commission [33]”; “institute (2005)” in L223 should be changed to “institute [34]”. And then, what does “(56)” in L430 means? On the other hand, we cannot find out “[54]” of the References list in the text of this manuscript.

Thank you for listing these. They have all been addressed in the text.

 4) Others: “and in Campylobacter” in L30-31 may be wrong in English; “liveweight, liveweight gain, mortality” in L273 should be deleted, because there are no data of those parameters in Table 6 and the results in mortality were described elsewhere (in L283-286); The authors said that the plant extract in this study increased lactic acid bacteria in L391-393, but I think that the bacteria were simply increased by dietary condition without amoxicillin from the results in this study.

Addressed and modified in the text.

 Lines of “Phosphorus”, “Crude Fat”, “Crude Fibre”, “Methionine”, “Cysteine”, “Histidine”, “Arginine”, “Isoleucine”, “Leucine”, “Phenylalanine”, “Tyrosine”, “Valine”, “Glycine”, “Serine” and “Phenylalanine + Tyrosine” in Table 2 should be omitted, because those nutrients are not important in this study. And then, Table 1 and 2 should be combined into one table.

The authors feel it is important to retain the nutritional profile of the plants within the paper as this data is extremely limited in the literature and it adds to the portfolio of knowledge on the plants. We acknowledge that the plants are not making a significant nutritional contribution and have made this clear in the text.

Round 2

Reviewer 2 Report

Thank you for your responses.  I made two comments in the text for you to add/correct. 

Author Response

Thank you again for your comments and suggestions. Please below the response to your comments.

Thank you for your responses.  I made two comments in the text for you to add/correct. I have included DMs in Table 2. 

I have included the name and location of the pharmaceutical company. Apologies for missing the degree celsius format. I have fixed this now.

Reviewer 3 Report

 I read the revised version of manuscript entitled “Effects of Agrimonia pilosa Ledeb, Anemone chinensis Bunge, and Smilax glabra Roxb on broiler performance, nutrient digestibility and gastrointestinal tract microorganisms” for possible publication in Animals. I think that the authors’ correspondence according to our comments are almost appropriate. But I have some further comments as follows:

  • I wonder that Fisher’s LSD test was used as a multiple range test for the comparison of means. The Fisher’s test must be available under limited conditions, such as same n numbers among treatments, and following of a normal distribution. I recommend using other multiple range test, such as tukey’s test. The authors should carry out statistical analyses again and revise the text of manuscript as necessary.
  • Others: “with [20]” in L116 should be changed to “with a list available online [20]”; “according to a commercial manual” should be inserted between “followed” and “[22]” in L130; The article of [39] should be deleted from the References list in L448-565, because of no citation in the text of manuscript.

Author Response

Thank you again for your comments and suggestions. Please below the response to your comments.

I wonder that Fisher’s LSD test was used as a multiple range test for the comparison of means. The Fisher’s test must be available under limited conditions, such as same n numbers among treatments, and following of a normal distribution. I recommend using other multiple range test, such as tukey’s test. The authors should carry out statistical analyses again and revise the text of manuscript as necessary.

We have considered this recommendation and feel that the Fisher’s LSD is the most appropriate, there was the same number of replications among treatments and the data followed a normal distribution. There are advantages and disadvantages to the various different post hoc tests and with the small number of treatments in this study, we think the Fisher’s LSD is most appropriate to test for treatment means.

Others: “with [20]” in L116 should be changed to “with a list available online [20]”; “according to a commercial manual” should be inserted between “followed” and “[22]” in L130; The article of [39] should be deleted from the References list in L448-565, because of no citation in the text of manuscript.

These changes have been made as recommended.